# Characterizing Garden Greenspace in a Medieval European City: Added Values of Spatial Resolution and Multi-Temporal Stereo Imagery

**Jingli Yan** [1,2,3], **Stijn Van der Linden** [4], **Yunyu Tian** [5], **Jo Van Valckenborgh** [4], **Veerle Strosse** [6] **and Ben Somers** [3,*]

1   School of Agriculture and Biology, Shanghai Jiao Tong University, Shanghai 200240, China; jingli.yan@sjtu.edu.cn
2   Shanghai Yangtze River Delta Eco-Environmental Change and Management Observation and Research Station, Ministry of Science and Technology, Shanghai 200240, China
3   Division of Forest, Nature and Landscape, KU Leuven, 200E Celestijnenlaan, 3001 Leuven, Belgium
4   Earth Observation Data Science, Digitaal Vlaanderen, 9000 Ghent, Belgium; stijn.vanderlinden@vlaanderen.be (S.V.d.L.); jo.vanvalckenborgh@vlaanderen.be (J.V.V.)
5   Laboratory of Geo-Information Science and Remote Sensing, Wageningen University & Research, 6708PB Wageningen, The Netherlands; yunyu.tian@wur.nl
6   Department of Environment, Flemish Government, 1000 Brussel, Belgium; veerle.strosse@vlaanderen.be
*   Correspondence: ben.somers@kuleuven.be

**Abstract:** Domestic gardens provide residents with immediate access to landscape amenities and numerous ecological provisions. These ecological provisions have been proven to be largely determined by greenspace composition and landscape, but the fragmentation and heterogeneity of garden environments present challenges to greenspace mapping. Here, we first developed a recognition method to create a garden parcel data set in the medieval Leuven city of Belgium, based on the land use layers and agricultural land parcels. Then, we applied multi-sourced satellite imagery to evaluate the added value of spatial resolution, plant phenology and 3D structure in identifying four vegetation types. Finally, we characterized the greenspace landscapes in garden parcels. Compared with single ALOS-2 imagery, SPOT-7 imagery and Pleiades-1A imagery increased the overall accuracy by 4% and 8%, respectively. The accuracy improvement (21%) produced from multi-temporal stereo Pleiades-1A imagery strongly verified the significance of plant phenology and 3D structure in garden mapping. The average greenspace cover in garden parcels was 71% but varied from 56% in urban gardens to 82% in rural gardens. The garden greenspace landscape is fragmented by the artificial structures in urban areas but has a more aggregated size and less complex shapes in rural areas. This study calls for greater attention to be paid to gardens, and for multi-disciplinary studies conducted in collaboration with urban ecologists and landscape designers to maximize the benefits to residents of both immediate landscape amenities and ecological provisions, in the face of global environmental changes and public health risks.

**Keywords:** domestic gardens; greenspace mapping; garden landscapes; multi-temporal stereo imagery; vegetation types

## 1. Introduction

Expanding cities worldwide have pledged to increase urban green infrastructure (UGI) [1] and maintain an extensive UGI network [2], as it provides numerous ecosystem services contributing to human well-being and sustainable urban development. Domestic gardens are the most promising places to carry out these long-term urban development plans [3], as they comprise more than one-third of the overall urban area in many cities [4,5]. However, domestic gardens have received comparatively less attention from scholars, likely due to the small parcel size and the lack of regulation. Furthermore, the majority of the general public is unaware of the environmental value of their own gardens and

how best to improve their ecological provisions. Homeowners' preferences in individual gardens increase the diversity and heterogeneity in pattern and landscape, which presents challenges to accurate greenspace mapping. Therefore, quantifying UGI composition and landscapes at the garden scale would be a valuable asset for understanding the current status of domestic gardens.

Factors relating to UGI composition, such as type and species, are imperative for ecological provision evaluation and rigorous ecological modeling. For instance, urban trees are more efficient in mitigating urban heat than grass [6,7]. Needle-leaf species generally capture and immobilize more air particles than broad-leaf species [8,9]. The significance of vegetation composition demands more powerful classification methods and data sets for UGI mapping, in particular for small-size domestic gardens. In recent decades, the increasing availability of very high-resolution satellite imagery and advanced remote sensing techniques has encouraged increasing numbers of attempts to use instant satellite imagery to deal with the heterogeneous UGI composition. Existing evidence verified that UGI mapping performance depends on study location [10,11], imagery properties [12,13], classification methods [14,15] and produced vegetation classes [16,17]. Mathieu et al. [16] combined IKONOS images and an object-based classification method to identify large-scale vegetation communities in urban areas of Dunedin City, New Zealand, and the overall accuracy ranged from 64% to 77%. Pu and Landry [12] produced an overall accuracy ranging from 66% to 67% when identifying six tree species/groups using WorldView-2 image in Tampa, Florida, USA. Li et al. [14] showed that a single worldview satellite image produces relatively low accuracies (45–83%) for five tree species in Beijing, China. These relatively low accuracies and erratic performances suggest the insufficiency of instant imagery in recognizing vegetation types, or even plant species.

Elvidge and Portigal [18] observed dramatic spectral changes in grasslands but slightly spectral changes in evergreen species from time-series AVIRIS images in 1990. Since then, multi-temporal satellite imagery reflecting plant phenology has been used to produce more accurate identification of vegetation types. Wolter et al. [19] applied a time-series of Landsat TM and MSS imagery acquired from 1980 to 1992 to identify forest types in northern Wisconsin, US. Hemmerling et al. [20] applied a Sentinel-2A/B time series for mapping 17 tree species at a regional scale in a temperate forest region of Central Europe. The majority of multi-temporal studies have focused on natural forests based on medium/low spatial resolution satellite imagery [21–23]. Meanwhile, Light Detection and Range (LiDAR) has been considered an effective solution to depict UGI structure and height. For example, Terryn et al. [24] identified five tree species using structural traits from terrestrial laser scanning in Oxford, UK, with a success rate of 80%. Dalponte et al. [25] investigated the classification setups of hyperspectral images, multispectral satellite images, and airborne LiDAR data for identifying seven tree species in mountain forests. Pu and Landry [26] evaluated the abilities of multi-temporal Pleiades satellite images with airborne LiDAR data for classifying seven urban tree species. However, the widespread applications of LiDAR have been largely limited by their availability and costs.

Essentially, multi-sourced data sets improve UGI mapping with their more abundant spatial and temporal information. The introduce of multi-sourced data set may increase uncertainties caused by systematic and non-systematic factors, whereas, with the development of remote sensed techniques, increasing satellites capture a greater variety of image products, like multi-temporal and stereoscopic images. These images from the same satellite platform reduced the uncertainties of combining applications. In addition, more effective image processing methods are also developed to deal with application issues that may raise uncertainties. Among remote sensed products, multi-temporal stereo imagery, which holds the advantages of being able to assess both plant phenology and 3D structure, provides great potential to facilitate UGI mapping [27–29]. While the general and public UGI have received intensive attention, UGI mapping in domestic gardens has been less studied [30,31]. Moreover, the land management practices of domestic gardens are diverse among stakeholders and lack regulation. Some quantitative studies have suggested a gen-

eral declining UGI trend in domestic gardens [32–34]. Therefore, the large-scale greenspace monitoring in domestic gardens will enable local authorities to assess the UGI benefits of urban domestic gardens.

Here, we first developed an approach to derive garden parcel data sets based on available spatial data sets, and then we demonstrated the UGI composition and landscape at garden scale by applying multi-temporal stereo imagery to UGI mapping in domestic gardens. During the UGI mapping, we developed classification schemes based on ALOS-2, SPOT-7 and Pleiades imagery. These were designed to (1) evaluate the added value of increased spatial resolution, plant phenology and 3D structure for identifying vegetation types, and (2) demonstrate the composition and landscape variations of domestic gardens in the compact European city of Leuven, Belgium. Our case study exemplified an application of garden monitoring and called for greater attention from both local authorities and stakeholders to garden management and ecological provisions.

## 2. Materials and Data Sets

### 2.1. Study Area

Leuven is an integrative part of the "Flemish Diamond", which locates in the heart of western Europe and consists of four agglomerations taking up four corners of an abstract diamond shape (Brussels-Ghent-Antwerp-Leuven). The urban domestic gardens can be given a highly important role in Flanders, and 73% of the houses have a domestic garden. Most areas of Flanders can be interpreted as urban sprawl filled with a maze of garden complexes. The greenspace in Leuven is mainly urban parks, agricultural lands and natural forests. The average greenspace coverage of domestic gardens is only 57% with an average size of 366 m$^2$. Although the total greenspace covers 65% of prefectural Leuven city (37.13 of 56.63 Km$^2$; Federal Ministry of Home Affairs, 2018), the greenspace is unevenly spatially distributed across the city. The city area outside the "ring road" is featured by lower building density but larger-size agricultural lands, whereas the compact urban area circled by the "ring road" is characterized by higher building density and smaller-size private garden mosaics (Figure 1). The developed urban area of Leuven has expanded eastwards, presenting an urban–rural continuum. Therefore, the small medieval town of Leuven with various functional zones and landscapes features would be an ideal place to evaluate greenspace mapping.

### 2.2. Remotely Sensed Imagery

ALOS-2 (Advanced Land Observation Satellite) imagery, SPOT-7 (Satellite pour l'Observation de la Terre) imagery and Pleiades-1A imagery (Table 1) were employed in this case study. ALOS-2 imagery (Figure 1a) acquired on 16 May 2019, and SPOT-7 imagery (Figure 1b) acquired on 22 July 2019. ALOS-2 senses the Earth's surface at a spatial resolution of 2.5 m for panchromatic and 10 m for multi-spectral (MS) bands. The SPOT-7 satellite can image the earth with a spatial resolution of 1.5 m at the panchromatic band and 6 m at the multi-spectral bands. Pleiades-1A satellite imagery (Figure 1c) acquired on 4 December 2019 (Figure 2a), 7 March 2020 (Figure 2b) and 31 July 2020 (Figure 2c,d) were also employed in this study. The Pleiades-1A satellite offers a spatial resolution of 0.5 m for the panchromatic (480–830 nm) and 2 m for the multispectral bands (Blue: 430–550 nm; Green: 490–610 nm; Red: 600–720nm; Near-Infrared: 750–950 nm). The summertime Pleiades-1A imagery (31 July 2020) is a stereo image, which provides almost simultaneous images from two different views (forward scanning in Figure 2c; backward scanning in Figure 2d) for the same area at the same spatial resolution. In general, a stereo-image (forward and backward looking) can produce a better accuracy of digital ground models in gentle terrain areas, while a tri-stereo-image (nadir, forward and backward looking) can be used for most terrains (Pleiades Imagery User Guide, 2020).

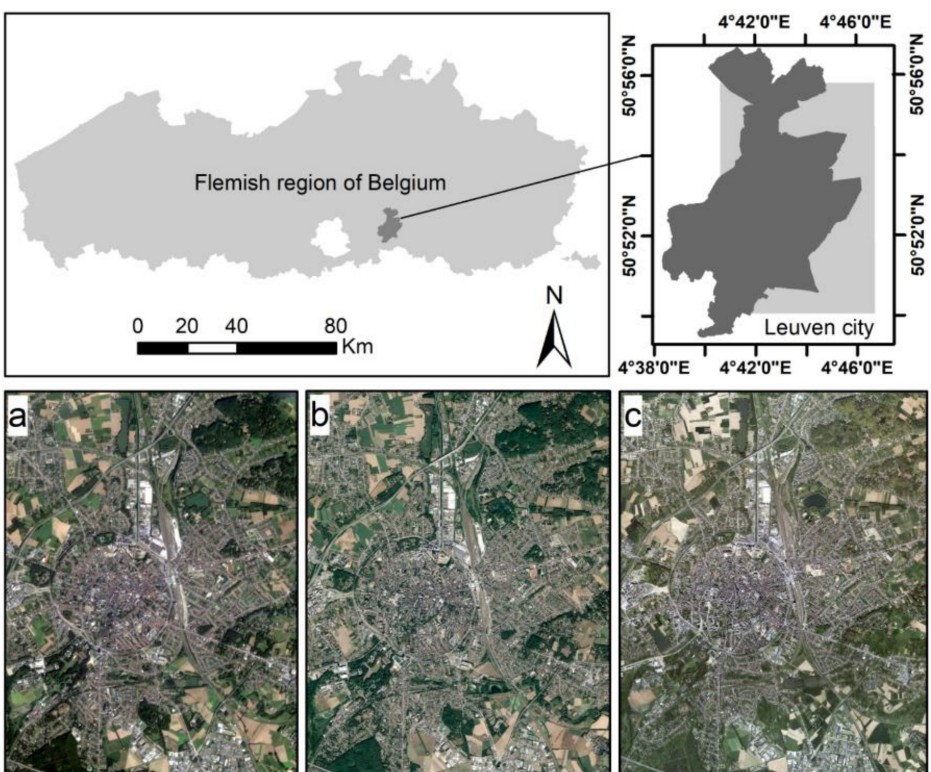

**Figure 1.** The study area demonstrated by satellite imagery in true color. (**a**) ALOS-2, (**b**) SPOT-7 imagery, (**c**) Pleiades-1A imagery.

**Table 1.** The basic features of satellite imagery used in classification.

| Satellite | Acquisition Date | Spatial Resolution | Multi-Spectral Band | Stereo Image |
|---|---|---|---|---|
| ALOS-2 | 16 May 2019 | 2.5 m | Blue: 420–500 nm; Green: 520–600 nm; Red: 610–690 nm; Near-Infrared: 760–890 nm | No |
| SPOT-7 | 22 July 2019 | 1.5 m | Blue: 455–525 nm; Green: 530–590 nm; Red: 625–695 nm; Near-Infrared: 760–890 nm | No |
| Pleiades-1A | 4 December 2019 | 0.5 m | Blue: 430–550 nm; Green: 490–610 nm; Red: 600–720 nm; Near Infrared: 750–950 nm | No |
| | 7 March 2020 | 0.5 m | | No |
| | 31 July 2020 | 0.5 m | | Yes |

These satellite imagery data were orthographic calibration-ready standard products obtained under cloudless conditions. A series of preprocessing steps were performed by the vendor (Airbus Defence and Space, 2020), including internal sensor geometry correction, removal of optical distortions and line-rate variations, and band registration. Afterward, we applied the Gramm–Schmidt algorithm [35] to pan-sharpening the multi-spectral bands, as the Gramm–Schmidt algorithm produces better spectral quality and maintains original spectrum of imagery [36]. Finally, we performed the geometric corrections by manually selecting more than 100 pairs of tie-points across Leuven city and controlling the root mean square under 0.5 using the aerial photography of summer 2018 as the reference map. The aerial photography was collected in the summer flying season during June to July of 2018,

and subsequently the production of an Orthophoto with a ground resolution of 25 cm (https://download.vlaanderen.be/, accessed on: 31 May 2020).

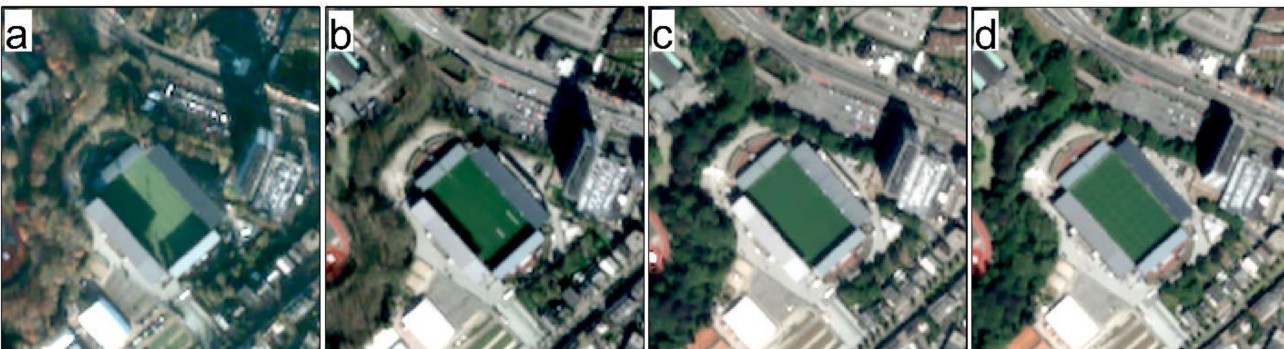

**Figure 2.** The examples of multi-temporal stereo Pleiades-1A satellite imagery. (**a**) winter image acquired in December 2019; (**b**) spring image acquired in March 2020; (**c**) summer image acquired at 10:58:23.4 of 31 July 2020 (along-track incidence of 15.41°); (**d**) summer image acquired at 10:59:10.6 of 31 July 2020 (along-track incidence of −10.22°).

### 2.3. Thematic Layers

#### 2.3.1. Plant Phenology (PP)

Multi-temporal imagery offers seasonal changes in plant physical and chemical characteristics. The variations of seasonal change among different plants contribute to the separation of vegetation types and even tree species. Evergreen plants are essentially unchanged in coloration and foliation across seasons, but most deciduous species show strong variation between the growing season and the non-growing season. In addition, plant phenological characteristics might vary greatly among functional types or plant species. For example, native populous trees usually sprout out in early spring and wither away in early fall, while the exotic populous trees introduced from warmer regions require a higher accumulated temperature to sprout out and fade away at a later time point. As the NDVI [37] is the most common index to reflect plant phenological changes, we calculated a plant phenology index (PP) based on NDVI values of multi-temporal Pleiades-1A imagery. In our case study, the non-growing season, growing season and the transition stage were represented by the imagery of December, July and March, respectively. For a ground object, if the NDVI of the December image (NDVI_Dec) was greater than that of the March image (NDVI_Mar), then PP = NDVI_Jul–NDVI_Dec. Otherwise, PP = NDVI_Jul–NDVI_Mar.

#### 2.3.2. Normalized Digital Surface Model (nDSM)

An nDSM represents the absolute height of objects above the ground surface, such as buildings and trees [38]. It can usually be created by subtracting a Digital Terrain Model (DTM) from a Digital Surface Model (DSM). A DSM represents the surface of the earth, including vegetation, buildings and other artificial features. A DTM only includes the elevation of the ground surface removing vegetation, buildings, and other artificial features, but roads and bridges are retained. Herein we generated an nDSM layer from stereo Pleiades-1A satellite imagery acquired in 31 July 2020.

The nDSM generation was performed in PCI Geomatica software (Geomatica Ortho-Engine). First, we selected the rational function for the optical satellite modeling method according to the data set type and platform. As the stereo satellite images are Orthoready standard products (ESPG:31370) and well-aligned to each other, we skipped the GCP/TP collection. Subsequently, we generated epipolar pairs between two stereoscopic images and automatically extracted the geocoded DSM (Figure 3a) using semi-global matching methods at a spatial resolution of 0.5 m. In PCI Geomatica, there are two primary tools to convert a DSM to a DTM (Figure 3c). The DEM editing tool is a manual method of conversion, while the DSM2DTM algorithm provides an automated capability. In this project,

we chose the manual DEM editing tool. During the conversion, several filter methods (terrain filters; bump and pit filters; median filters; clamp filters; manual touch-up) can be repeatedly applied by order until we produce a satisfying result.

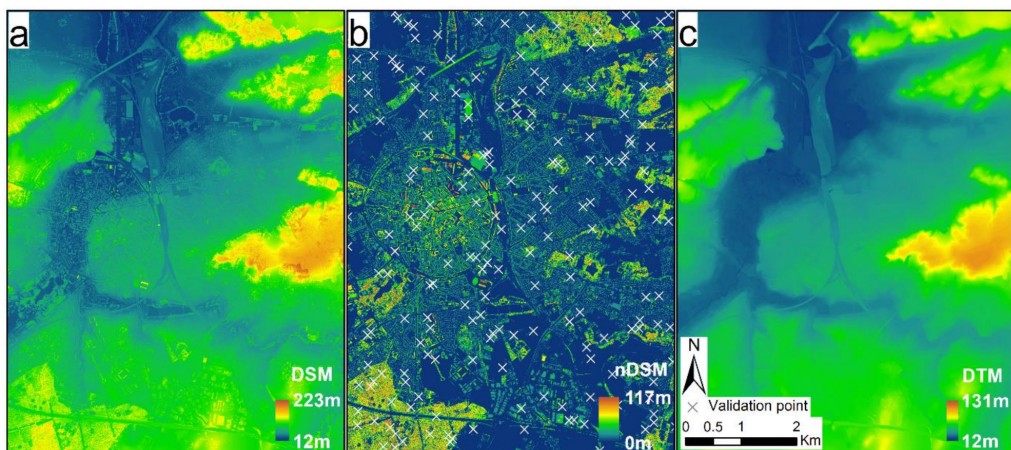

**Figure 3.** The digital models produced from stereo Pleiades-1A satellite imagery. (**a**) Digital Surface Model (DSM); (**b**) normalized Digital Surface Model (nDSM); (**c**) Digital Terrain Model (DTM).

### 2.3.3. Garden Parcels

Informatie Vlaanderen Agency (EODaS) developed a workflow to automatically delineate garden parcels from official governmental spatial data layers. Taking into account various garden definitions [30,39], we defined the 'garden parcel' in the present analysis as places that are covered with cultivated greenspace to provide enjoyment. Therefore, a garden can be residential (e.g., backyards), public (e.g., city parks), industrial (e.g., parking lots) or semi-public greenspace (e.g., hospitals). Following this definition, the garden parcels were modified based on the large-scale reference file (GRB, https://overheid.vlaanderen.be/GRB-Wat-is-het-GRB, accessed on: 31 May 2020) and the Agricultural Land Parcel Registration (LBPC, https://lv.vlaanderen.be/en, accessed on: 31 May 2020). The GRB file contains geographical and characteristic information of well-definable, conventionally accepted reference data, including buildings, administrative parcels, roads, waterways, railways and so on. The LBPC contains parcels in agricultural use for the Department of Agriculture and Fisheries to fulfill their mission: 'contribution to the development of future-oriented agricultural and fisheries policies and quality service to the Flemish agro-food sector'. A garden parcel in our case study is an administrative parcel that

1. does not overlap with an agricultural parcel or a railroad;
2. contains at least one main buildings;
3. or contains one or more side buildings, which share a border with an administrative parcel that contains one or more main buildings;
4. or does not contain a building but overlaps with a building block that contains more than 60% administrative parcels with one or more main buildings;
5. has been cut by roads, main buildings, and buildings > 20 m$^2$;
6. has been cleaned from slivers.
7. The above procedures produced more than 15,000 garden parcels in study area.

### 2.3.4. Field Inventory

To obtain inclusive ground samples, we conducted extensive field campaigns in various local climate zones (LCZs) (Table 2) over the entirety of Leuven city. The classification scheme of LCZs was adopted from Stewart and Oke [40], which was based on (i) building height, (ii) building density and (iii) greenspace coverage at an analytical unit of 100 m × 100 m grid cell (Table 2). In summer 2019, 203 gardens across Leuven city (20–30 gardens for each LCZ depending on voluntary applicants, Figure 4a) were selected

and investigated. In the field (Figure 4b), field experts manually drew all ground objects in a garden parcel on a printed copy of an aerial photograph, and recorded the land use type for each object, e.g., trees, shrubs, grass, farm plantings, buildings, roads, water, etc. All objects were digitalized into vector layers in QGIS (version 3.10) based on the field campaigns (Figure 4c). Finally, 3624 ground objects from 203 gardens were collected, and 3081 of them were vegetated objects.

**Table 2.** The classification of LCZs, * class is absent in study area.

| Building Density | Building Height | Greenspace Coverage | LCZ Class |
|---|---|---|---|
| ≥0.25 | ≥8 m | ≥67% | 111 * |
| | | 34−67% | 112 |
| | | ≤34% | 113 |
| | <8 m | ≥67% | 121 * |
| | | 34−67% | 122 |
| | | ≤34% | 123 |
| <0.25 | ≥8 m | ≥67% | 211 * |
| | | 34%−67% | 212 |
| | | ≤34% | 213 |
| | <8 m | ≥67% | 221 |
| | | 34−67% | 222 |
| | | ≤34% | 213 |

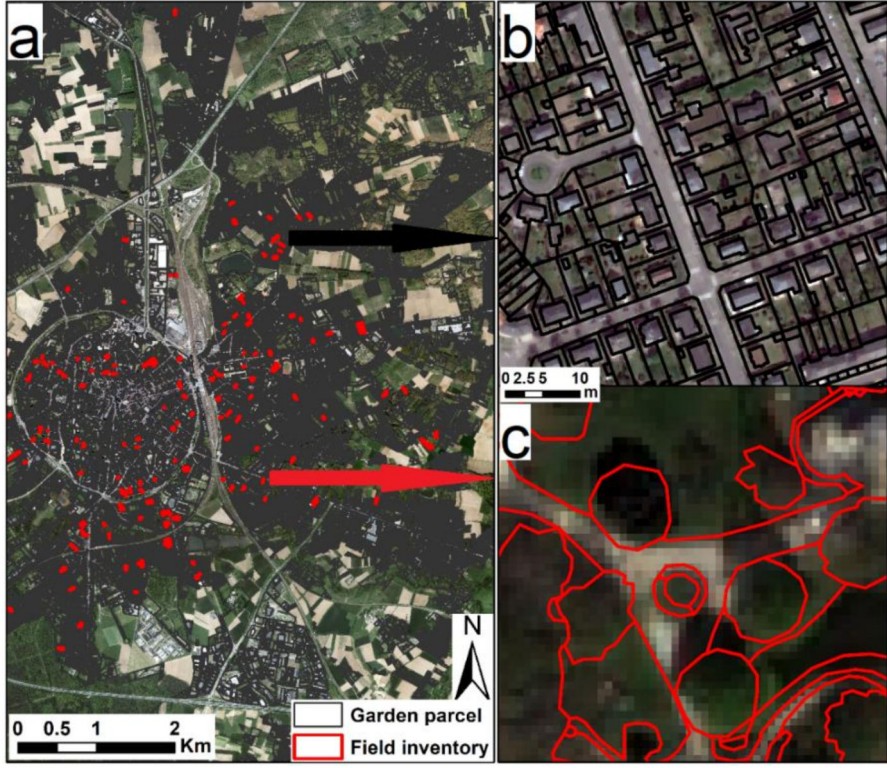

**Figure 4.** The garden objects in the field inventory. (**a**) distribution of investigated gardens; (**b**) garden parcels; (**c**) manually delineated garden objects.

## 3. Greenspace Mapping in Gardens

### 3.1. Classification Designs

This case study involved six classification schemes to evaluate the contribution of spatial resolution, plant phenology and 3D structure to UGI monitoring (Figure 5). First, we conducted a classification based on ALOS-2 imagery (referred to as scheme a), SPOT-

7 imagery (referred to as scheme b) and summer Pleiades-1A imagery (referred to as scheme c), to examine the mapping capacity of instant satellite imagery. Subsequently, we performed further classification schemes by introducing seasonal Pleiades-1A imagery (referred to as scheme d) and stereo Pleiades-1A imagery (referred to as scheme e) to scheme c. For scheme d, the plant phenology (PP) derived from multi-temporal Pleiades imagery was applied to separate deciduous and evergreen plants, while for scheme e, the nDSM derived from stereo Pleiades imagery was applied to identify plant height. For scheme e, we introduced both plant phenology and nDSM to scheme c (referred to as scheme f).

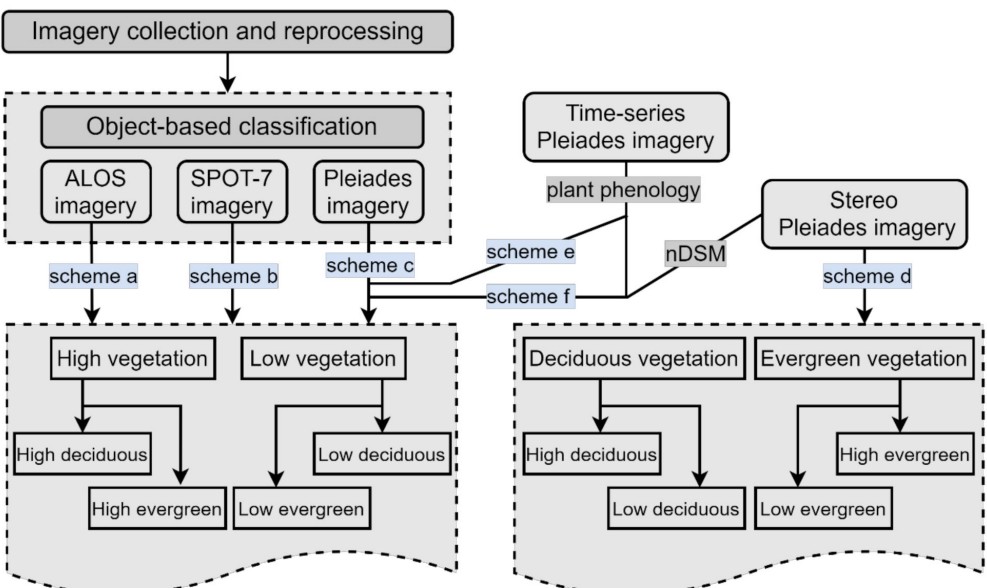

**Figure 5.** The conceptual diagram of classification designs. scheme a-Classification using ALOS-2 imagery; scheme b-Classification using SPOT-7 imagery; scheme c-Classification using Pleiades-1A imagery; scheme d-Classification using stereo Pleiades-1A imagery; scheme e-Classification using multi-temporal Pleiades-1A imagery; scheme f-Classification using multi-temporal stereo Pleiades-1A imagery.

An Object-Based Image Analysis (OBIA) [41,42] was conducted to produce four vegetation functional types: High Deciduous (HD), Low Deciduous (LD), High Evergreen (HE) and Low Evergreen (LE). OBIA can accurately delineate ground object boundaries and introduces many more spectral and spatial features than the pixel-based approach, especially for very high spatial resolution remotely sensed imagery [43]. An OBIA normally includes three procedures: image segmentation, sample training and classifier application. In present classifications, a multi-resolution segmentation (MRS) algorithm was applied to produce image objects at hierarchical levels. The ground samples were trained using Classification and Regression Tree (CART) to generate features and thresholds for separating vegetation types. A ruleset classification based on CART results was established and applied to produce a greenspace map.

### 3.2. Image Segmentation

Generating accurate image segments as identical as the ground-truth objects is a crucial precondition for a high-quality OBIA. The MRS algorithm embedded in eCognition Developer was applied to the segmentation. MRS is a bottom-up segmentation which consecutively merges pixels or existing image objects into bigger objects based on a relative homogeneity criterion [41]. The homogeneity criterion measures how homogeneous an image object is. Three parameters (scale, shape and compactness) can be modified to control homogeneity criterion [42]. A larger scale parameter possibly causes "under-segmentation" through generating image objects with mixed land covers, whereas a smaller scale parameter leads to "over-segmentation" of a fragmented landscape. Following the

scale parameter, the color and shape were modified to refine the shape of the segments. The value of the shape, which equals one minus that of the color, determines how much weight (0–1) was placed on the shape when generating segmented objects. The parameter compactness is a shape-related factor, which determines the compactness or smoothness of segmented objects.

Compared with the commonly used visual inspection or the "trial and error approach", the Estimation of Scale Parameter (ESP) [44] is more efficient and objective. ESP produces multiple segmentations of the same image by constantly increasing the scale parameter and calculating local variance at the scene level for each scale to identify that which produces the highest spatial independence of segmented objects (Figure 6). In one ESP practice, several scale parameters can be indicated for different land covers, e.g., greenspace, water and buildings. Concerning the comparability of the final classification results, we have to produce identical image objects for all the six different schemes (Figure 5). We conducted the MRS based on all 20 multi-spectral bands of satellite imagery in Table 1. These multi-spectral bands were equally weighted in segmentation, and the produced image objects were communal for classification schemes. ESP results suggest three potential scale parameters (25, 55 and 90; Figure 6) targeting different classes: scale 25 for greenspace, 55 for vegetation types and 90 for water and impervious surfaces. Most previous studies found that more meaningful objects would be extracted with a higher weight for the color criterion [45]. Therefore, we produced three-level hierarchical image objects with 25, 55 and 90 as the scale parameters, 0.2 as the shape and 0.5 as the compactness.

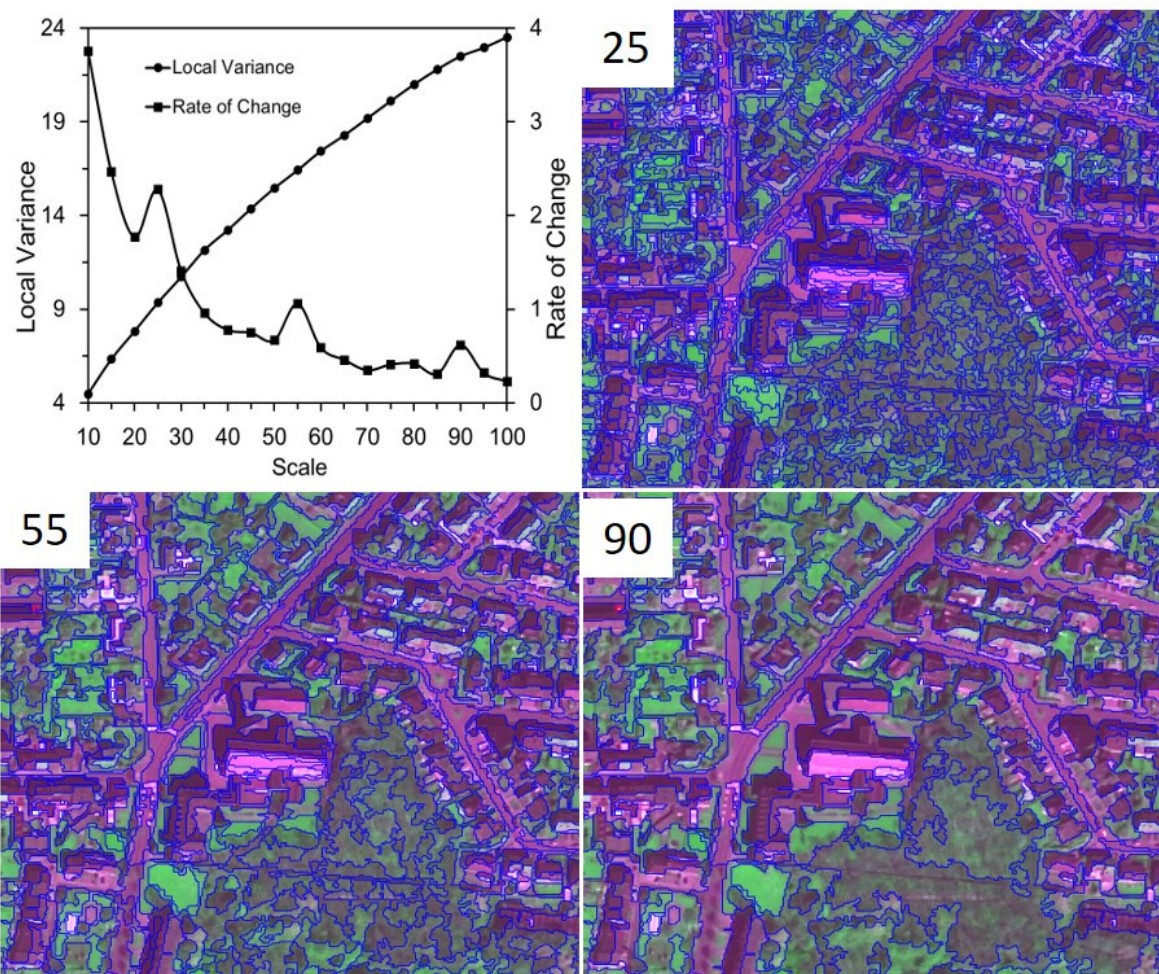

**Figure 6.** The results of ESP and produced image objects.

*3.3. Classification Procedures*

All segmented image objects were firstly separated into shaded objects and unshaded objects using thresholds of mean brightness and mean red band (Table 3). Subsequently, the vegetated objects, indicated by NDVI and mean blue band, were separately extracted from unshaded and shaded objects. Here, we applied the same features but different thresholds to recognize vegetated objects among unshaded and shaded objects. For instance, the NDVI threshold for vegetated objects in unshaded objects was 0.24, while in unshaded objects was 0.13 (Table 3). For recognized vegetation objects, different classification procedures among classification schemes were subsequently performed to separation of functional types (HD, LD, HE, LE).

3.3.1. Classification Using Single Satellite Imagery (Schemes a to c)

For vegetated objects on single ALOS-2, SPOT-7 and Pleiades imagery, we first applied the Tree-Grass Difference Index (TDGI, expression = $-$Log(canny)+Brightness) [46] to separate high vegetation (trees and shrub) and low vegetation (grass). Subsequently, CART produced features and thresholds to differentiate deciduous and evergreen in high vegetation and low vegetation, respectively. CART is a non-parametric classification method which makes no assumptions regarding the underlying distribution of the predictor variables and identifies splitting variables based on an exhaustive search of all possibilities [47]. CART usually produces straightforward rules and clear information on the importance of features. For a classification with few land cover classes, the CART only needs limited computational effort but produces reasonable accuracy [48]. Two-thirds of garden vegetation samples were randomly selected as training samples and were ex-ported into CART to produce decision trees for separating vegetation types (Figure 7). For shaded areas, we divided them into vegetation shadow (shadow caused by trees and shrubs) and building shadow (shadow caused by buildings), as the two types of shadow are too distinct in spectrum and shape characteristics to apply the same thresholds when identifying vegetation types under them. For instance, building shadow is generally darker and larger than vegetation shadow. Therefore, the shaded objects with an area greater than 400 pixels (100 m$^2$) and brightness lower than 24 were classified as building shadow, and the rest were vegetation shadow. In addition, the more boundaries of an object shared with vegetated objects, the higher the possibility that the object is vegetation. Then, shaded vegetation was recognized from the shadow by combining NDVI (0.13 for tree shadow, 0.08 for building shadow) with a border relative to vegetation objects (0.5). Afterward, the vegetation types in shaded objects were separated using the same features as in unshaded vegetation but with different thresholds (Table 3).

3.3.2. Classification Integrating Multi-Temporal Stereo Satellite Imagery (Schemes d to f)

With respect to classification schemes d and e, Plant phenology (from seasonal Pleiades imagery) and nDSM (from stereo Pleiades imagery) were separately introduced into summer Pleiades-1A imagery classification (scheme c) as an auxiliary layer. For scheme d, we first divided the vegetated objects into three groups (deciduous, mixed vegetation, evergreen) according to PP values. Vegetated objects were classified as evergreen when PP < 0.17 for unshaded objects (0.12 for shaded objects), and as deciduous when PP > 0.25 for unshaded objects (0.21 for shaded objects). Vegetated objects with 0.17 < PP < 0.25 for unshaded objects (0.12 < PP < 0.21 for shaded objects) were classified as mixed vegetation and were further classified into deciduous and evergreen vegetation using the features and thresholds listed in Table 3. In addition, the obtained deciduous and evergreen vegetation was further classified as high vegetation or low vegetation using TDGI. In contrast, high vegetation and low vegetation were first identified by nDSM at a break point of 5 m in scheme e and then separated into deciduous and evergreen vegetation using the features and thresholds listed in Table 3. For scheme f, we applied nDSM to separate high and low vegetation, and PP to separate deciduous and evergreen using the features and thresholds in scheme d and e to ensure the validity of classification comparisons.

**Table 3.** Features and thresholds for classifications.

| Class Name (Object Level) | Feature (Threshold) | Class Name (Object Level) | Feature (Threshold) |
|---|---|---|---|
| Unshaded area (25) | Brightness (36); Mean Red (387) | Shaded area (25) | Brightness (36); Mean Red (387) |
| | | Tree & Building shadow (55) | Area (100 m$^2$); Brightness (24) Relative border to green (0.5) |
| Unshaded green (25) | NDVI (0.24); Mean Blue (445) | Shaded green (55) | NDVI (0.13) in tree shadow NDVI (0.08) in building show |
| High & low green (55) | $-$Log(canny_NIR) + Brightness (51.24) | High & low green (55) | $-$Log(canny_NIR)+Brightness (36.81) |
| Deciduous & evergreen (55) | Mean NIR (236); Ratio G/R(1.34); Hue-R_G_B (0.21) | Deciduous & evergreen (55) | Mean NIR (197); Ratio G/R (1.15); Hue-R_G_B (0.16) |
| High deciduous & evergreen (55) | Ratio G/R (1.52); GLCM-H_NIR (0.14); GLCM-H_G (0.19); Hue-R_G_B (0.31) | High deciduous & evergreen (55) | Ratio G/R (1.29); GLCM-H_NIR (0.06); GLCM-H_G (0.12); Hue-R_G_B (0.22) |
| Low deciduous & evergreen (55) | Mean NIR (279); Ratio R/NIR (0.56); Hue-G_R_NIR (0.23); GLCM-H_NIR (0.1) | Low deciduous & evergreen (55) | Mean NIR (234); Ratio R/NIR (0.46); Hue-G_R_NIR (0.14); GLCM-H_NIR (0.08) |
| Plant Phenology (55) | 0.17; 0.25 | Plant phenology (55) | 0.12; 0.21 |
| nDSM (55) | 5 m | nDSM (55) | 5 m |

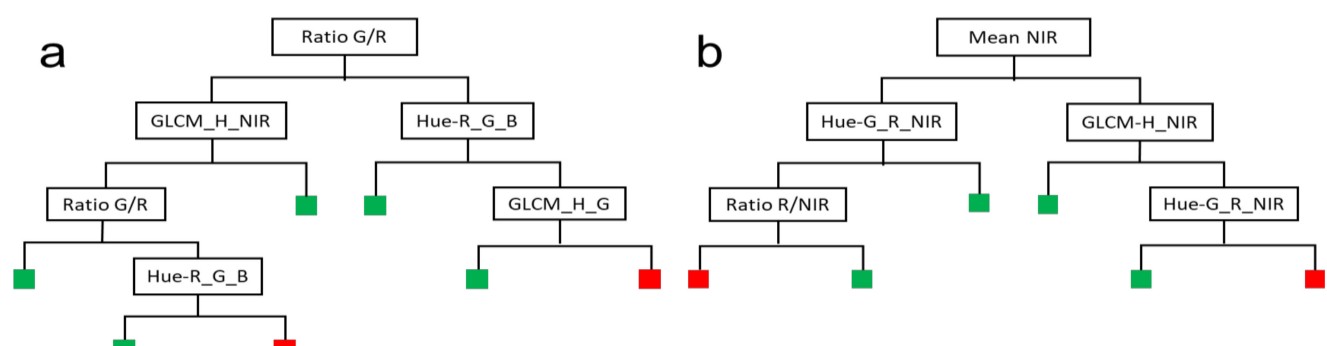

**Figure 7.** The decision trees resulted from CART. (**a**) Decision tree for high deciduous and high evergreen; (**b**) Decision tree for low deciduous and low evergreen. The green square indicates the targeting class, while the red square indicates the other class.

### 3.4. Accuracy Assessment

We applied two validation methods, "point validation" and "polygon validation", to assess the classification schemes. The "point validation" was conducted to confirm the classification accuracy on a ground point. We performed point validation on 800 validation sites randomly selected from the whole study area. We then generated their confusion matrixes and calculated Kappa coefficients for vegetation functional types. In contrast, the "polygon validation" was performed at garden scale within the 203 gardens in which conducted the field investigations and manual delineations. We compared the percentage coverage of both garden greenspace and vegetation types in a garden between produced greenspace map and the greenspace objects observed during the garden survey. The polygon validation was performed at three spatial scales: statistical sector, building block, and garden parcel. A statistical sector of Belgium (2019 version) is the basic territorial unit resulting from the subdivision of the territory of municipalities for the dissemination of its statistics at a finer level than the municipal level (accessed online by https://download.vlaanderen.be/, accessed on 31 May 2020). A city block is the smallest group of buildings that is surrounded by streets, not counting any type of thoroughfare within the area of a building or comparable object. The garden parcel map was developed by EODoS as described in Section 2.3.

## 4. Results and Discussion

### 4.1. Validation of Thematic Layers

#### 4.1.1. nDSM Layer

To evaluate the nDSM layer generated from stereo Pleiades-1A imagery (referred to as nDSM_P), we here introduced an nDSM derived from LiDAR altitude data (Digital Altitude Model Flanders II, DHMV II, https://overheid.vlaanderen.be/dhm-dhmv-ii-brondata, accessed on 31 May 2020) as the reference map (referred to as nDSM_R). The DHMV II was collected in summer 2015 by Aerodata Surveys Neder-land BV (https://www.geobusiness.nl/leden/aerodata, accessed on 31 May 2020) featuring an average point density of 15 points/m². The processing procedures in LAStools software include the detection and removal of noisy returns, the detection of ground returns, the creation of DTM through interpolation, the derivation of all non-ground returns and the creation of a CHM by extracting the maximum height for each cell. More processing details can be found in Degerickx et al. [49]. A total of 200 validation points were randomly selected across the study area (Figure 3b). The height from nDSM layers showed a clear linear relationship ($R^2$ = 0.976, Figure 8a). The nDSM produced from stereo Pleiades-1A imagery overestimated the object height by 0.92m on average (Figure 8b).

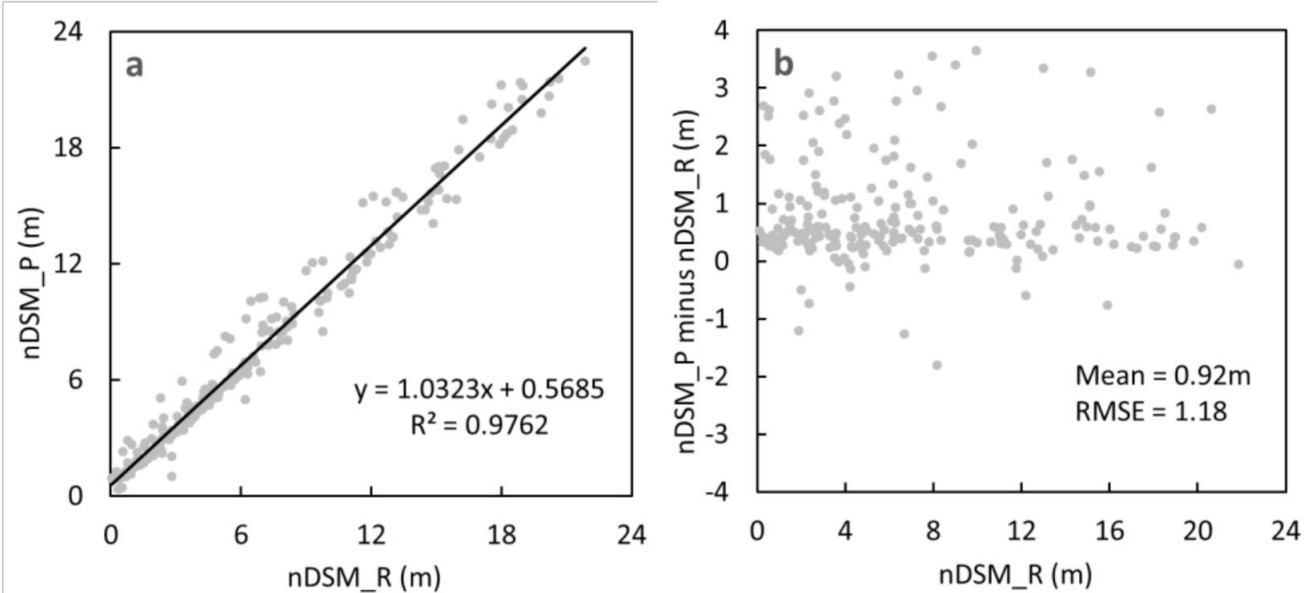

**Figure 8.** The nDSM validation. (**a**) comparison of nDSM layers from stereo Pleiades satellite imagery (nDSM_P) and LiDAR (nDSM_R); (**b**) the differences between two nDSM layers.

#### 4.1.2. Garden Parcels

The garden parcel map in Section 2.3.3 was evaluated by 400 randomly selected garden parcels, 200 garden parcels and 200 non-garden parcels. The overall accuracy of the garden parcel map was 90%. The misclassified gardens were generally agricultural lands. We further categorized all gardens parcels into three groups (urban gardens; suburban gardens; and exurban gardens) by equally dividing the value range of building density at the statistical sector scale (Figure 9a). The building density of a corresponding statistical sector is the ratio of the overall floor area of buildings to the area of the Belgian statistical sector where the garden parcel is located (Figure 9b). A garden parcel (Figure 9c) was classified as urban garden if the corresponding statistical sector had higher building densities, and likewise for suburban garden parcels and exurban garden parcels.

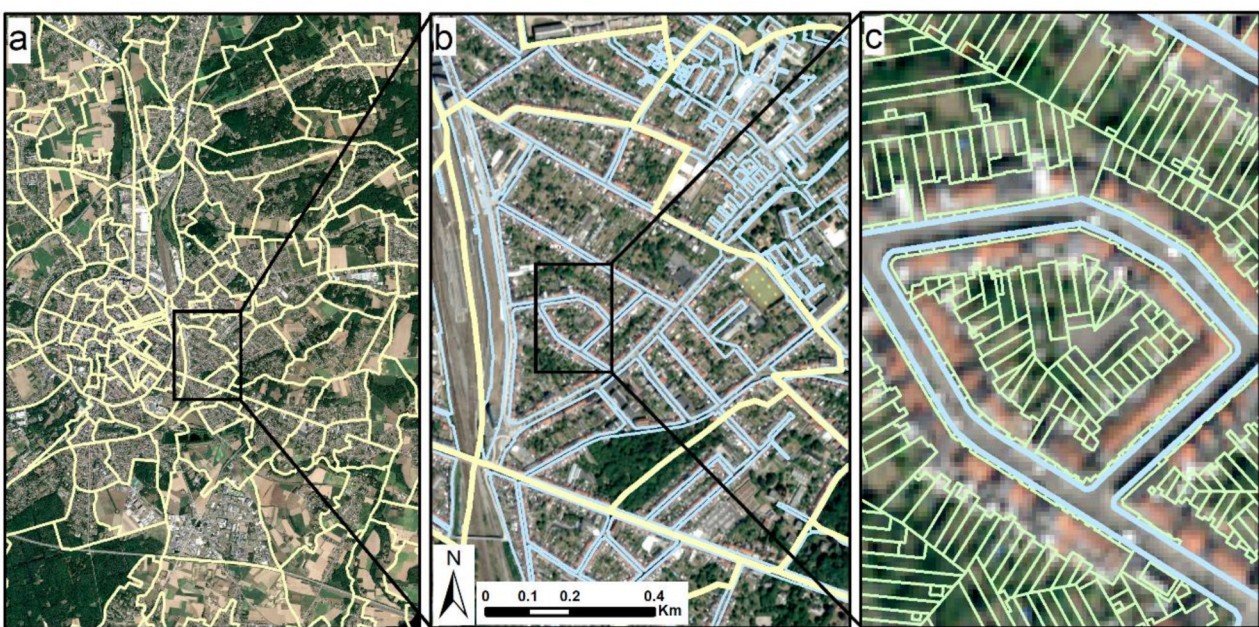

**Figure 9.** The spatial scale of study area: (**a**) statistical sector; (**b**) building block; (**c**) garden parcel.

### 4.1.3. Image Objects

No segmentation result is fully convincing if it does not satisfy human eyes. Therefore, we evaluated the image segments through comparison with round-truth objects. We randomly selected 100 image objects and overlapped the image segments with manually delineated objects (Figure 10). Statistics suggested that the image segments were likely to be over-segmented for the patches over 2500 m$^2$ (Figure 11a) and under-segmented when the patch size was less than 2500 m$^2$ (Figure 11b). In addition, the image segments obtained similar sizes to the delineated objects, with an average overlap ratio of 91.75 $\pm$ 4.12% (Figure 11c). Considering that the image objects were produced on the basis of 20 multi-spectral bands of used satellite imagery, the reasonable accuracies verified the accurate segmentation and provided a reliable basis for the following classification procedures.

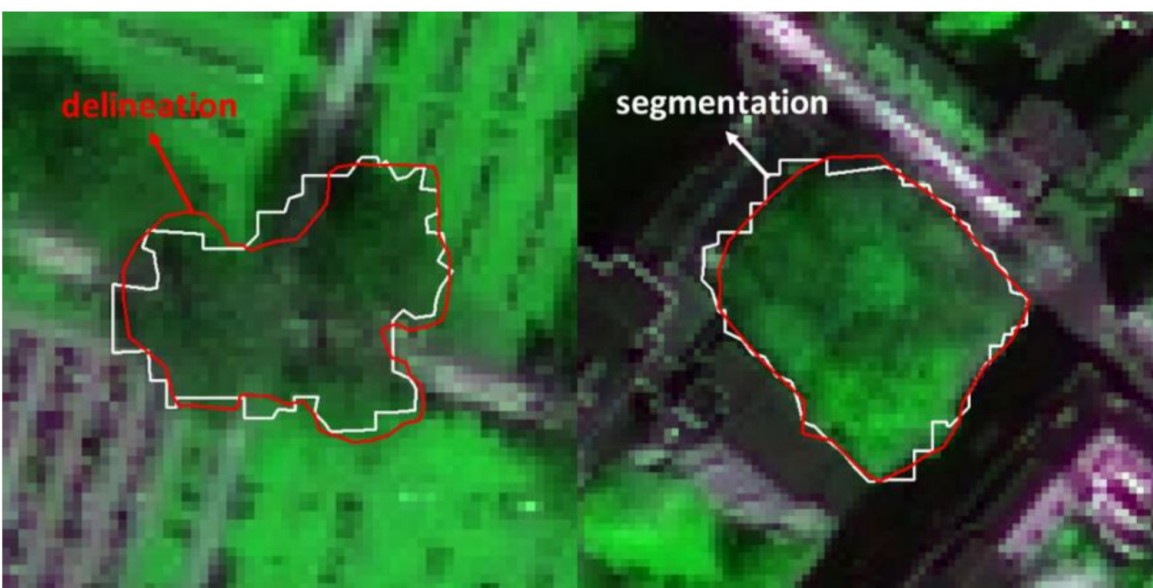

**Figure 10.** The comparisons of image objects produced by manual delineations and object-based segmentations.

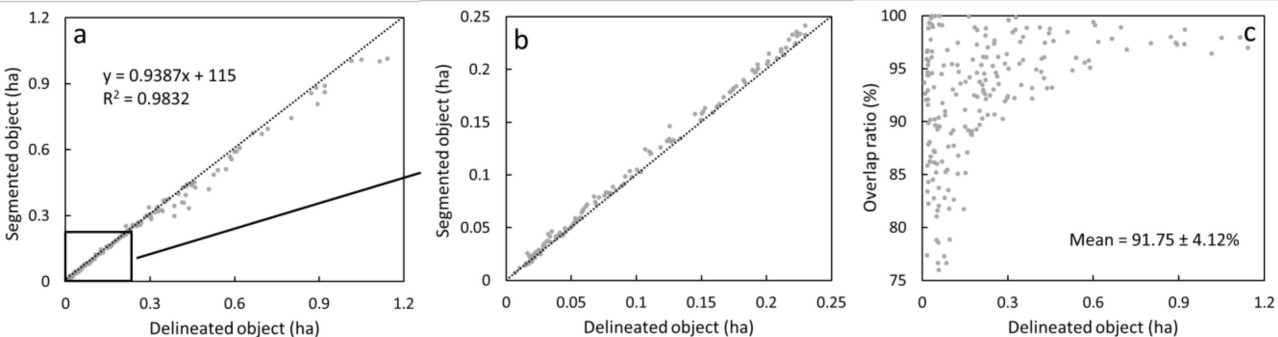

**Figure 11.** The differences between segmented and manually delineated objects. (**a**) 100 random selected objects; (**b**) the image objects less than 2500 m$^2$; (**c**) the overlap percentages of selected objects; the gray dash line is 1:1 line.

### 4.2. A Higher Spatial Resolution Improves Greenspace Mapping in Gardens

The overall accuracy of UGI mapping in domestic gardens (Figure 12; Table 4) increased from 71.13% for ALOS-2 imagery (scheme a) to 76.38% for SPOT-7 imagery (scheme b), and further to 79.25% for Pleiades-1A imagery (scheme c). The remotely sensed classifications identified 7–12% (12.06% for ALOS imagery to 7.63% for Pleiades imagery) less greenspace coverage than in situ investigations. These improvements in overall accuracy (4.25% vs. 3.87%) and coverage differences (2.24% vs. 2.09%) verified the contributions of the spatial resolution of satellite imagery. Similar positive effects are supported by existing studies [10,50,51]. Li et al. [14] suggested that the overall accuracy from WorldView-3 imagery (1.2 m for MS band) was 2–4% higher than that from WorldView-2 imagery (2 m for MS band) when identifying tree species in Beijing urban areas. Pu and Landry [12] showed a 3–7% increment in the overall accuracy of urban tree species/groups when applying IKONOS imagery (4m for MS bands) and WorldView-2 imagery (2 m for MS band). Together with existing studies, we have exemplified that the very high spatial resolution of remote sensing products is a precondition for fine-scale UGI mapping, although it may depend on study regions, classification approaches, etc. [15,52].

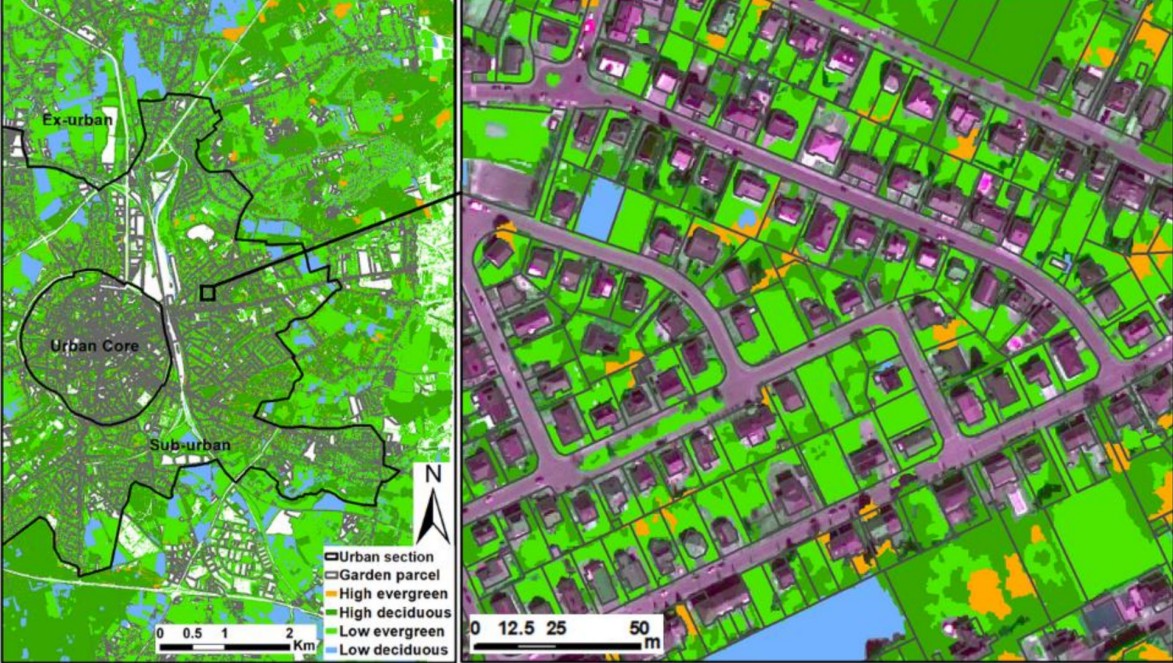

**Figure 12.** The classification result from multi-temporal stereo Pleiades-1A imagery.

**Table 4.** The confusion matrixes of greenspace maps.

| | | Scheme a- ALOS-2 Imagery | | | | | | Scheme b- SPOT-7 Imagery | | | | | | Scheme c- Pleiades-1A Imagery | | | | | |
|---|---|---|---|---|---|---|---|---|---|---|---|---|---|---|---|---|---|---|---|
| OA | | 71.13 | | | | | | 75.38 | | | | | | 79.25 | | | | | |
| Kappa | | 0.634 | | | | | | 0.688 | | | | | | 0.735 | | | | | |
| | | Reference | | | | | | | | | | | | | | | | | |
| | | HD | HE | LD | LE | NV | UA | HD | HE | LD | LE | NV | UA | HD | HE | LD | LE | NV | UA |
| Classified | HD | **152** | 11 | 19 | 16 | 7 | 74.15 | **163** | 11 | 15 | 11 | 8 | 78.37 | **171** | 8 | 13 | 12 | 7 | 81.04 |
| | HE | 17 | **60** | 13 | 10 | 3 | 58.25 | 13 | **69** | 14 | 9 | 4 | 63.3 | 11 | **73** | 11 | 7 | 2 | 70.19 |
| | LD | 22 | 9 | **82** | 13 | 5 | 62.6 | 19 | 7 | **88** | 12 | 6 | 66.67 | 15 | 9 | **97** | 9 | 5 | 71.85 |
| | LE | 16 | 12 | 12 | **131** | 5 | 74.29 | 15 | 8 | 10 | **138** | 1 | 80.23 | 13 | 7 | 7 | **145** | 3 | 82.86 |
| | OT | 14 | 9 | 10 | 8 | **144** | 77.84 | 11 | 6 | 9 | 8 | **145** | 81.01 | 11 | 4 | 8 | 5 | **147** | 84.57 |
| | PA | 68.78 | 59.41 | 60.29 | 73.6 | 87.8 | | 73.76 | 68.32 | 64.71 | 77.53 | 88.41 | | 77.38 | 72.28 | 71.32 | 81.46 | 90.24 | |

| | | Scheme d- Multi-Temporal Pleiades Imagery | | | | | | Scheme e- Stereo Pleiades Imagery | | | | | | Scheme f- Multi-Temporal Stereo Pleiades Imagery | | | | | |
|---|---|---|---|---|---|---|---|---|---|---|---|---|---|---|---|---|---|---|---|
| OA | | 84.5 | | | | | | 86.13 | | | | | | 92.75 | | | | | |
| Kappa | | 0.803 | | | | | | 0.822 | | | | | | 0.908 | | | | | |
| | | Reference | | | | | | | | | | | | | | | | | |
| | | HD | HE | LD | LE | **NV** | UA | HD | HE | LD | LE | **NV** | UA | HD | HE | LD | LE | **NV** | UA |
| Classified | HD | **184** | 5 | 6 | 11 | 6 | 86.79 | **188** | 12 | 6 | 7 | 4 | 87.50 | **203** | 3 | 6 | 3 | 3 | 93.12 |
| | HE | 7 | **81** | 10 | 5 | 1 | 77.88 | 15 | **84** | 3 | 3 | 1 | 80.77 | 7 | **95** | 2 | 2 | 0 | 89.62 |
| | LD | 15 | 6 | **107** | 4 | 3 | 79.26 | 5 | 2 | **111** | 13 | 4 | 83.46 | 5 | 1 | **121** | 4 | 2 | 90.98 |
| | LE | 9 | 4 | 8 | **150** | 0 | 87.72 | 6 | 3 | 12 | **152** | 2 | 87.36 | 3 | 1 | 5 | **166** | 2 | 93.79 |
| | OT | 6 | 5 | 5 | 8 | **154** | 86.52 | 8 | 3 | 3 | 6 | **153** | 88.44 | 3 | 1 | 2 | 3 | **157** | 94.58 |
| | PA | 83.26 | 80.20 | 78.68 | 84.27 | 93.90 | | 85.52 | 83.17 | 81.62 | 85.39 | 93.29 | | 91.86 | 94.06 | 88.97 | 93.26 | 95.73 | |

OA—Overall Accuracy; UA—User's Accuracy; PA—Producer's Accuracy; HD—Higher Deciduous; HE—High Evergreen; LD—Low Deciduous; LE—Low Evergreen; NV—Non-Vegetation.

High evergreen obtained the lowest accuracy, while low evergreen (slightly better than high deciduous) obtained the highest accuracy, as suggested by the confusion matrixes (Table 4). For classifications based on single satellite imagery (schemes a–c), the misclassified confusions generally present a nearly random pattern among vegetation types. Compared with field investigation, remotely sensed recognition underestimated high deciduous vegetation by 14.18% to 18.55% and low deciduous by 22.43% to 25.19%, but it overestimated low evergreen vegetation by 12.79% to 15.12% and high evergreen by 20.09% to 23.36% (Table 5). These results indicated that single satellite imagery has limited capacity in capturing either plant morphology or height structure in urban environments.

**Table 5.** The average percentage of the cover difference between produced greenspace map and the observed greenspace in 203 investigated gardens during the field campaign.

| Greenspace Type | Scheme a | Scheme b | Scheme c | Scheme d | Scheme e | Scheme f |
|---|---|---|---|---|---|---|
| Greenspace | $-12.06 \pm 5.25$ | $-9.72 \pm 4.47$ | $-7.63 \pm 4.36$ | $-5.44 \pm 2.58$ | $-5.29 \pm 2.45$ | $-3.53 \pm 2.12$ |
| High Deciduous | $-18.55 \pm 7.31$ | $-16.87 \pm 5.52$ | $-14.18 \pm 5.89$ | $-11.32 \pm 4.57$ | $-10.15 \pm 3.96$ | $-7.06 \pm 2.85$ |
| High Evergreen | $23.36 \pm 8.39$ | $20.08 \pm 7.12$ | $17.57 \pm 7.24$ | $14.34 \pm 5.63$ | $13.48 \pm 5.02$ | $10.63 \pm 3.79$ |
| Low Deciduous | $-25.19 \pm 9.75$ | $-22.43 \pm 7.65$ | $-19.83 \pm 8.09$ | $-15.92 \pm 5.18$ | $-15.36 \pm 4.97$ | $-11.19 \pm 4.01$ |
| Low Evergreen | $15.12 \pm 6.36$ | $12.79 \pm 5.21$ | $11.37 \pm 4.74$ | $8.16 \pm 3.86$ | $7.53 \pm 3.03$ | $5.22 \pm 2.17$ |

Aside from spatial resolution, imaging properties also impact UGI mapping, especially in enclosed places such as domestic gardens. First, the capturing swath width (20 km for Pleiades satellite, 60 km for SPOT-7 satellite, and 70 km for ALOS satellite) is related to shadow coverage and texture. The fragmented and heterogeneous urban environments mostly consist of interlaced built-up and greenspace mosaics. Leuven is a medieval old town with high compactness which significantly increases the challenges of UGI mapping. Consequently, the shadow treatment of satellite imagery is critical for urban land cover mapping, as these shaded areas significantly minimize the spectral signature of land cover [53,54]. Second, the Pleiades satellite covers more panchromatic and multi-temporal bands (in wavelength) than the other two, especially for the vegetation-sensitive red band and near-infrared band. Similarly, Pu and Landry [12] demonstrated that the additional four multi-spectral bands (especially the red edge band) of WorldView-2 imagery can improve the average accuracy by 6–9% for recognizing six tree species/groups in Florida, US, therefore proving that very high-resolution imagery is a premise for accurate urban mapping. Compared with SPOT-7 and ALOS-2 imagery, Pleiades-1A imagery has a clearer

view from orbit to ground surface, which produces fewer shaded areas but a more abundant spectrum, eventually improving UGI mapping using Pleiades imagery.

*4.3. Time-Series and Stereo Imagery Improve Greenspace Mapping in Gardens*

Compared with single Pleiades imagery (scheme c), multi-temporal imagery (scheme d) and stereo imagery (scheme e) significantly improved the classification accuracy by 5.25% and 6.88%, respectively. The accuracy was improved up to 13% when combining multi-temporal and stereo Pleiades imagery (scheme f). These improvements indicated that plant phenology and nDSM are a remarkable addition to single satellite imagery for UGI mapping. Moreover, the stereo imagery was slightly more influential than multi-temporal imagery (86.13% vs. 84.5%) in recognizing vegetation types (Table 4). The slightly lower accuracy from multi-temporal imagery was possibly caused by the desynchronized phenology of plant species and the blurred boundaries of spectral reflectance between evergreen and deciduous plants.

In UGI mappings, the nDSM is generally applied to separate plant heights. Its performance mostly depends on the spatial resolution of the original data source, e.g., point cloud density. Zhou and Qiu [55] applied WorldView-2 imagery and waveform LiDAR to identify trees and grass in Dallas, US, and found that the accuracy produced from fused data was 7–8% higher than that from WorldView-2 imagery. Liu et al. [56] concluded that LiDAR contributed 10% more accuracy than hyperspectral imagery in tree species recognition and achieved a nearly 20% higher accuracy when combining LiDAR and hyperspectral imagery in the greater Vancouver area, Canada. Currently, satellites are becoming increasingly capable of imaging (tri-)stereo images, but the majority of existing nDSM layers are still produced from LiDAR. In this study, the nDSM generated from stereo Pleiades-1A satellite imagery showed an equivalent efficacy to nDSM produced from a LiDAR source ($R^2$ = 0.98, RMSE = 1.18). This may partly be because the flat terrain of the study area is favored by stereo imagery, but the greater ability of tri-stereo imagery in dealing with various terrains would be a relief in the absence of the LiDAR data set.

Differing from nDSM, the performance of plant phenology is largely determined by synergistic effects, e.g., satellite platform, spatial resolution, observation time, and also frequency. The deciduous species generally start to wither and decolorize in the late fall or early winter, which leads to yellow reflections after the reduction in chlorophyll and the increment in cancroid within plant leaves [45]. Some evergreen plants de-colorize during the non-growing season, but to a much lesser degree than deciduous plants. Therefore, the observation time of satellite imagery is a vital feature to discriminate deciduous and evergreen species [57]. Tigges et al. [52] showed that a larger number of sensed images contribute to more accurate landcover dynamic observations. In our study, the three Pleiades-1A satellite images acquired from March, July and December might fail to capture phenological changes completely, as January is possibly the optimal timing point for the study area to discriminate deciduous and evergreen plants.

In contrast to the randomly distributed confusions from classifications using single imagery, classifications integrating multi-sourced imagery presented aggregated confusion patterns. A large fraction of misclassifications occurred between high and low vegetation when using multi-temporal imagery, but confusions often occurred between deciduous and evergreen when using stereo imagery (Table 4). Evergreen species were always "over-classified" while deciduous species were "under-classified" (Table 5). These confusing patterns were the result of multiple effects. Low deciduous (5.21%) was the least covered vegetation type, followed by high evergreen, with a coverage of 6.93% in the study area. Moreover, these vegetation types were fragmented and scattered, and spectrally affected by the surrounding environment, especially in enclosed gardens. Furthermore, the compact medieval city of Leuven contains much shading and many obscured areas. The evergreen species are typically pine and fir trees, generally dark in color, which limits the availability of spectral and textural features. Therefore, for these vegetation types, it is difficult to

collect high-level training samples and sufficient spectral signatures to meet the requisite of rational recognition accuracy.

### 4.4. Greenspace Landscapes in Gardens Parcels

The study area had more than 15,000 garden parcels (as defined in Section 2.3.3) and covered 30.85% of the study area (24.13 of 78.21 Km$^2$, Figures 4 and 12) with an average size of 474 m$^2$. This forms the landscape view, in that the large area of urban sprawl is filled with a maze of garden mosaics. The garden parcels contain 18.82 Km$^2$ of greenspace and much more vegetation than the other land use types (77.95% vs. 46.32%) across the study area. The average greenspace coverage was 51.05% in statistical sectors, 31.48% in building blocks, and 70.98% in garden parcels (Figure 13; Table 6). This variable coverage at different scales implies that the domestic gardens contribute the majority of urban greenspace [16,30]. At the broad scale of the Flemish region of Belgium, approximately 73% of houses have a domestic garden [58], which take up 8.3% of the total area. The garden coverage of 30.85% in our study area makes sense, because the "diamond region" is the most populated region in Belgium.

Global observations revealed that, in populated cities, UGI landscapes are generally small-sized and inlaid in a matrix of artificial structures in the urban core area but are more aggregated in rural areas [59]. Although medieval Leuven is a small city by international standards, its compactness results in a similar greenspace landscape to populated cities, which implies a demand for smart greening strategies [60]. For example, the exurban gardens (1017.83 m$^2$) are significantly larger than the other gardens (637.97 m$^2$ of suburban gardens, and 258.59 m$^2$ of urban gardens), and the greenspace within gardens decreased in mean patch size (153.48 to 21.35 m$^2$) but increased in patch density (65.09 to 425.32/ha). Specifically, the townhouses and city apartments of Leuven city are usually associated with small backyards, which constrains the patch size of garden greenspace but increases fragmentation and shape complexity [61]. In contrast, detached houses and single-family houses in rural areas generally contain a large amount of vegetated area, which would aggrandize greenspace coverage and mean patch size (Table 6).

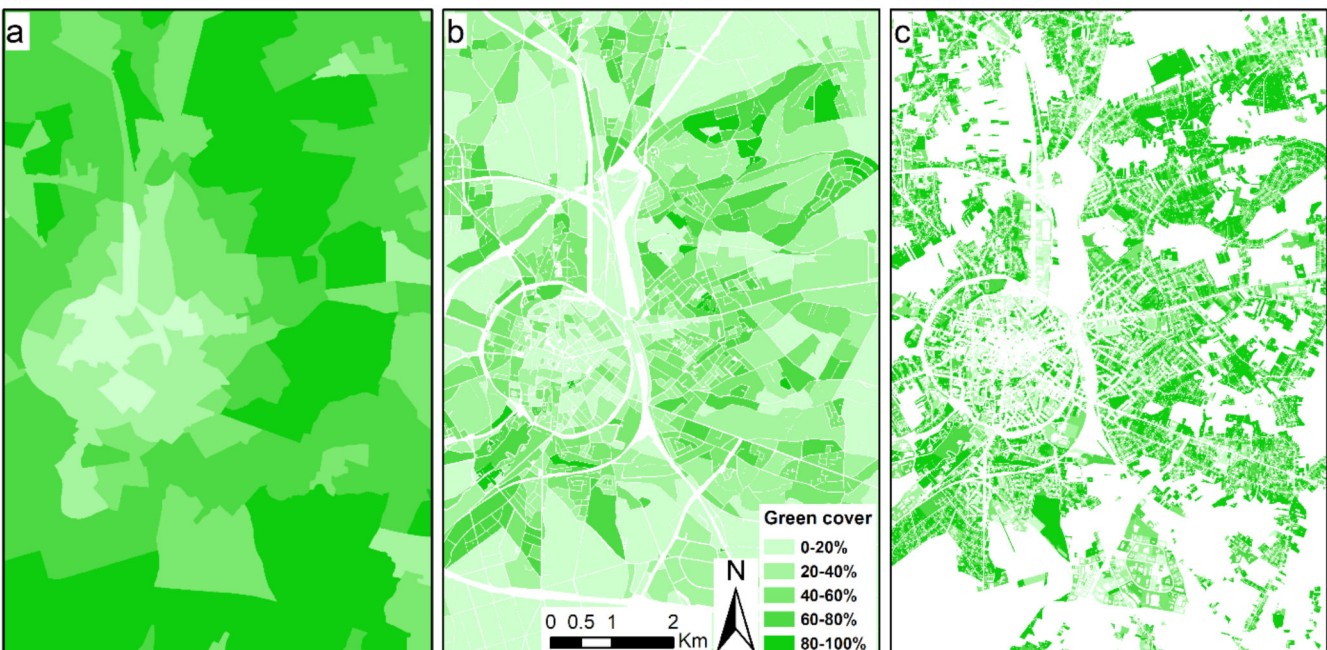

**Figure 13.** The spatial variations of greenspace at scales. (**a**) Statistical sector; (**b**) building block; (**c**) garden parcel. Figures share the legend.

**Table 6.** The landscape characteristics of garden greenspace.

| | | TA (ha) | PC (%) | PD (/ha) | ED (/ha) | LPI (%) | MPS (m²) |
|---|---|---|---|---|---|---|---|
| Garden | Green space | 1881.74 | 70.98 | 227.86 | 1402.33 | 78.45 | 92.93 |
| | *High evergreen* | *174.31* | *6.93* | *102.91* | *441.43* | *7.27* | *12.24* |
| | *High deciduous* | *694.12* | *23.46* | *637.28* | *1860.27* | *26.42* | *29.81* |
| | *Low evergreen* | *859.97* | *35.38* | *298.03* | *1103.72* | *40.13* | *55.58* |
| | *Low deciduous* | *153.34* | *5.21* | *79.45* | *386.09* | *4.63* | *17.11* |
| Urban garden | Green space | 122.03 | 56.46 | 425.32 | 2049.57 | 60.69 | 21.35 |
| | *High evergreen* | *19.14* | *8.56* | *152.28* | *461.68* | *7.55* | *8.72* |
| | *High deciduous* | *47.65* | *20.43* | *855.94* | *2387.14* | *23.61* | *15.19* |
| | *Low evergreen* | *42.84* | *22.29* | *573.6* | *1778.39* | *25.83* | *23.08* |
| | *Low deciduous* | *12.40* | *5.18* | *130.46* | *424.91* | *3.74* | *7.43* |
| Suburban garden | Green space | 926.18 | 70.85 | 146.83 | 1236.06 | 75.41 | 86.16 |
| | *High evergreen* | *89.46* | *7.75* | *108.43* | *463.04* | *6.48* | *11.36* |
| | *High deciduous* | *318.59* | *25.79* | *542.74* | *1296.98* | *25.35* | *25.42* |
| | *Low evergreen* | *441.11* | *31.51* | *314.56* | *982.95* | *38.52* | *44.91* |
| | *Low deciduous* | *77.02* | *5.8* | *89.77* | *372.63* | *5.06* | *14.28* |
| Exurban garden | Green space | 833.54 | 82.43 | 65.09 | 591.15 | 94.37 | 153.48 |
| | *High evergreen* | *67.93* | *6.67* | *59.58* | *437.28* | *7.96* | *16.7* |
| | *High deciduous* | *295.54* | *29.41* | *251.34* | *883.51* | *29.13* | *41.05* |
| | *Low evergreen* | *406.62* | *40.12* | *115.67* | *601.73* | *51.47* | *73.19* |
| | *Low deciduous* | *63.45* | *6.23* | *50.02* | *359.42* | *5.85* | *23.96* |

Existing urban ecological studies have revealed that greenspace landscapes (i.e., compositions and configurations) could provide numerous ecosystem services contributing to urban sustainability and human well-being. For example, trees were more efficient in mitigating urban heat islands than grass [7]. Nevertheless, low evergreen (grassland) was the most vegetated type (35.34%) across studied gardens, which makes these gardens more vulnerable to urban heat risks or other environmental hazards than the city parks containing many woods. In addition, the greenspace coverage, patch size and shape complexity are also positively related to their cooling effects [6]. In study area, the urban heat intensity is 2.8–4.7 °C and the discomfort index is 3.4–4.3 higher in urban areas than in rural areas during the summer. However, the greenspace in downtown gardens (6925 parcels) constitutes only a small fraction of total garden greenspace (1.22 of 18.82 Km² at an average coverage of 56.46%), whereas rural gardens (9128 parcels) provide a considerable proportion of total garden greenspace, 8.33 of 18.82 Km² at an average coverage of 82.43%. This striking contrast leads to large imbalances in ecosystem provisions and demands between urban gardens and rural areas. Therefore, it is worth emphasizing the pressing needs of interdisciplinary studies conducted in collaboration with landscape designers and urban ecologists. Such studies would provide better insight for landscape designers when they are balancing ecological functions and landscape amenities in garden regulations.

*4.5. Applicability and Limitation*

We valued the applicability very much when designing the methodology in the first place. A methodology derived from one place generally are difficult to transfer to other places of the world. There are several factors may improve the applicability of our study. First, the study area is a typical region, that represents heterogeneous garden environment and abundant vegetation resources, to test the methodology. Second, we selected the widely used

satellite imagery that has potentially contributions to our classification. Third, we evaluated the capacity of stereo-imagery in constructing 3D structure, concerning the less availability and accessibility of LiDAR data set in many regions. Finally, we present a straightforward methodology design, which is easy to understand, including segmentation validation, feature selection and classifier application. Those designs facilitated the applicability and replication of our analysis in other cities and regions.

Our case study exemplified the application of multi-sourced satellite imagery in quantifying composition and landscape in the less concerned gardens. This study could be improved in following aspects: (a) the plant phenologies in study area vary among vegetation types and species, so a bigger number of temporal images (in present we only have 3 images) can offer more variations of plant phenology; (b) the Pleiades imagery used in our study are stereoscopic imagery, a tri-stereoscopic imagery would produce more accurate 3D structure than stereo-image, in particular the north-western part of study region are hilly areas; and (c) we chose CART as the classification method, but many other classification methods like SVM, RF and CNN also produce reliable performances. Further studies may compare these classification methods to identify the most potential one for UGI mapping in domestic gardens.

## 5. Conclusions

Domestic gardens comprise more than one-third of urban areas in many cities but have been the focus of less study due to their small size and lack of regulation. Here, we developed an approach to derive garden parcels and evaluated the added values of multi-sourced satellite imagery in garden mapping in the medieval European city. Our results have proved the significant contributions of increased spatial resolution to vegetation type recognition in domestic gardens. The improvements were generally related to their spatial resolution increments. Application of multi-temporal stereo Pleiades-1A imagery improved the overall accuracy by 13% and greenspace coverage by 8.53% contrasting to classification of single Pleiades-1A imagery. The garden greenspace covers 30.85% of total garden parcels, and the average greenspace coverage is 70.98%. Along with the urban-rural continuum, the average garden size increased from 258.59 $m^2$ for urban gardens to 1017.83 $m^2$ for rural gardens, while the average greenspace coverage increased from 56.46% to 82.43%. Additionally, the aggregated greenspace mosaics in rural gardens indicated by the largest patch index led to the increased mean patch size and reduced greenspace fragmentation and shape complexity.

City residents are less educated with regard to the ecological provisions of urban gardens, and how they could improve their function. Studies such as this one call for multidisciplinary collaborations between urban ecologists, landscape designers, and the local community. These studies and collaborations offer residents an opportunity to understand the environmental value of their gardens and enlighten decision makers with regard to greenspace planning to optimize ecosystem services and landscape amenities.

**Author Contributions:** Conceptualization, J.Y., J.V.V. and B.S.; methodology, J.Y., B.S. and Y.T.; formal analysis, J.Y. and Y.T.; resources, J.V.V., V.S. and B.S.; data curation, J.Y. and S.V.d.L.; writing—original draft preparation, J.Y. and Y.T.; writing—review and editing, J.V.V. and B.S.; visualization, J.Y. and S.V.d.L.; supervision, B.S.; project administration, V.S.; funding acquisition, J.V.V. and B.S. All authors have read and agreed to the published version of the manuscript.

**Funding:** This research was funded by "Belgian Federal Science Policy Office" in the framework of the STEREOIII program (projects GARMON (grant number: SR/00/363) and GARVAL (grant number: SR/01/201)).

**Institutional Review Board Statement:** Not applicable.

**Informed Consent Statement:** Not applicable.

**Data Availability Statement:** The data sets would be available upon reasonable requests.

**Conflicts of Interest:** The authors declare no conflict of interest.

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
