# Peer review of "Characterizing Garden Greenspace in a Medieval European City: Added Values of Spatial Resolution and Multi-Temporal Stereo Imagery"

_remotesensing, doi:10.3390/rs14051169_

Round 1

Reviewer 1 Report

Overall comment

This study explores how the use of multi-sourced satellite imagery is able to increase the accuracy of the classification of garden greenspace. It is generally an interesting topic. Please find below for detailed comments for the authors to consider.

Detailed comments:

  1. As stated in the study (Line 96-107), one key goal is to evaluate the added-values of increased spatial resolution, plant phenology, and three-dimensional structure for identifying vegetation types. However, many of the recent studies have tried to take these factors into account and found the benefits in terms of increasing the accuracy in classifying the greenspace. So what’s the main originality of this study? It would be better to highlight the solved gaps/originality of this study explicitly.
  2. In order to better highlight the originality or the significance of this study, I think a systematic literature review would be helpful (to summarize the limitations and gaps in previous studies).
  3. For another main purpose of this study, which is to demonstrate the composition and landscape variations of residential gardens of a city, the results are mainly descriptive, using some landscape characteristics for different types of greenspace in terms of different types of garden (shown in Table 5). Such results are preliminary and descriptive. The authors may consider to try more spatial modelling or spatial analysis based on these results (like to justify the possible reasons/drivers causing the composition variation, or etc.), in order to generate more useful findings and implications.

Reviewer 2 Report

This manuscript shows a comparison of different data for greenspace mapping. The aim and purpose is good. The main problem is that the method lacks of novelty, making the paper is more like a report. More features usually lead to a better classification result. A more important problem is how to make full utilization of these data and features. The current method is a quite traditional OBIA method, the fusion of data is a simple stack of layers, and the fusion of features is based on the object-level summaries. The parameters of CART (e.g., the size of the tree, the minimum number of splits) are even not tuned and optimized. Given so many advanced classification methods such as CNN and RNN, with the former has advantages in extracting high-level features and the latter is good at processing seasonal images, the method in this manuscript is out of date and lack of comparison. Therefore, the results of the accuracies appeared in Abstract and Conclusion are not persuasive since the improvements in accuracy definitely change when applying a different classification method, or based on a different set of samples.

When assessing the accuracy of garden parcels, the error caused by segmentation and classification should be separated.

Reviewer 3 Report

In general, the paper is interesting and well written, there are comments need to be addressed before the paper can be found acceptable.

  1. Please explain why the use of multitemporal and multisource data can essentially improve the classification or mapping accuracy? Since the introduce of such data may also increase the uncertainty due to the data acquisition quality.
  2. The following very relevant papers can be also added into the Introduction part, which is about multitemporal image classification and multisource image fusion for classification.
  • Novel Cross-Resolution Feature-Level Fusion for Joint Classification of Multispectral and Panchromatic Remote Sensing Images, DOI: 10.1109/TGRS.2021.3127710.
  • A Novel Multitemporal Deep Fusion Network (MDFN) for Short-Term Multitemporal HR Images Classification, DOI:1109/JSTARS.2021.3119942.
  • A Novel Feature Fusion Approach for VHR Remote Sensing Image Classification, DOI: 10.1109/JSTARS.2020.3041868
  1. The performance of OBIA scheme is usually limited by the segmentation scale, please explain more clearly the setting of parameters in your experiments, and are these parameter values are suitable for different land-covers and different scenarios?
  2. The CART is a good classification method, however, from the performance point of view, there are many good machine learning techniques such as SVM, random forest, and more advanced deep learning methods. Please clarify the main motivation that you choose CART in your work, and how is the potential to use other excellent classifier.
  3. Did you consider the different spatial resolution and spectral bands information in ALOS, SPOT and Pleiades-1A that may effect on the classification results?
  4. The legend in Figure 12 is somehow misleading, which should be associated to all (a)-(c) subfigures, not only to (b).

Reviewer 4 Report

This paper aims to “Characterizing Garden greenspace in a medieval European city: added values of spatial resolution and multi-temporal stereo imagery”. The topic is interesting; I would like to suggest minor revisions for this study.  

  1. It is better to provide metadata of the remote sensing images as a table.
  2. What are the limitations of the study?
  3. How about the applicability of the proposed methodology for other areas? Better to explain it.

Round 2

Reviewer 1 Report

I accept the response and revisions made by the authors. Proofreading is suggested in order to enhance the readability of the paper.

Author Response

Thanks so much for your valuable comments. We agree that our methodology design and results presentation have potential to be improved. We will keep that in mind in our future research.

Reviewer 2 Report

The authors response all the concerns.

Author Response

Thanks very much for your insightful comments in the review process.